# Using old antibiotics to treat ancient bacterium—β-lactams for *Bacillus anthracis* meningitis

**Assa Sittner, Amir Ben-Shmuel, Itai Glinert, Elad Bar-David, Josef Schlomovitz, David Kobiler, Shay Weiss, Haim Levy** [ORCID] *

Department of Infectious Diseases, Israel Institute for Biological Research, Ness Ziona, Israel

* haiml@iibr.gov.il

## Abstract

As *Bacillus anthracis* spores pose a proven bio-terror risk, the treatment focus has shifted from exposed populations to anthrax patients and the need for effective antibiotic treatment protocols increases. The CDC recommends carbapenems and Linezolid (oxazolidinone), for the treatment of anthrax, particularly for the late, meningeal stages of the disease. Previously we demonstrated that treatment with Meropenem or Linezolid, either as a single treatment or in combination with Ciprofloxacin, fails to protect rabbits from anthrax-meningitis. In addition, we showed that the failure of Meropenem was due to slow BBB penetration rather than low antibacterial activity. Herein, we tested the effect of increasing the dose of the antibiotic on treatment efficacy. We found that for full protection (88% cure rate) the dose should be increased four-fold from 40 mg/kg to 150 mg/kg. In addition, *B. anthracis* is a genetically stable bacterium and naturally occurring multidrug resistant *B. anthracis* strains have not been reported. In this manuscript, we report the efficacy of classical β-lactams as a single treatment or in combination with β-lactamase inhibitors in treating anthrax meningitis. We demonstrate that Ampicillin based treatment of anthrax meningitis is largely efficient (66%). The high efficacy (88–100%) of Augmentin (Amoxicillin and Clavulonic acid) and Unasyn (Ampicillin and Sulbactam) makes them a favorable choice due to reports of β-lactam resistant *B. anthracis* strains. Tazocin (Piperacillin and Tazobactam) proved inefficient compared to the highly efficient Augmentin and Unasyn.

## Introduction

*Bacillus anthracis*, the etiological cause of Anthrax, is a gram-positive spore forming bacterium. Anthrax is a zoonotic disease that usually afflicts herbivores and is usually transferred to humans by contact with contaminated animal products [1, 2]. The infecting form of *B. anthracis* is the spore that can infect the host via three major routes; skin, gastrointestinal tract and the respiratory system [3]. The route of infection determines the type of disease and therefore the outcome of the infection. Contact of spores with damaged skin will cause a local lesion (eschar), that in the absence of treatment will remain local and resolve spontaneously in 70–

**Data Availability Statement:** All relevant data are within the paper and its Supporting Information files

**Funding:** The authors received no specific funding for this work

**Competing interests:** The authors have declared that no competing interests exist.

80% of the cases [1, 4]. The remainder 20–30% of patients will progress to systemic infection that, in the absence of treatment, will end in death. Both, the gastrointestinal infection, resulting from the consumption of contaminated meat causing gut inflammation [5] and inhalational anthrax, which is the result of spore inhalation, become systemic in almost 100% of the cases and are almost always lethal without treatment [3]. While skin and gastrointestinal anthrax can be contracted naturally, inhalational anthrax is extremely rare and was considered an occupational disease mainly in workers and goat hair processing mills [6].

Unfortunately in the case of a massive spore discharge as a bio-weapon or bio-terror attack, the exposed population will be at risk of developing deadly inhalational anthrax [7]. Inhalational anthrax is a two phase disease where the first stage resembles the common flu or atypical pneumonia followed by a short acute phase leading to death [3]. The symptomatic phase is most probably the result of systemic bacterial spread (bacteremia) and the acute phase includes in most cases infection of the central nervous system (CNS) in the form of hemorrhage and meningitis [8–13]. As the risk of the use of *B. anthracis* spores for bio-terror arose, the focus of treatment has shifted from exposed populations to anthrax patients with the need for effective antibiotic treatment protocols. Historically, treatment of anthrax was based on penicillin G [14].With the approval of new antibiotics, treatment recommendations shifted to tetracyclines and fluoroquinolones i.e. Doxycycline and Ciprofloxacin [3, 15].

During the Amerithrax mail *B*. anthrais spore attacks, Ciprofloxacin was the major antibiotic that was prescribed as post exposure prophylactics (PEP) for population at risk of exposure [16]. This prophylactic treatment was highly effective since there were no reports of systemic anthrax in individuals treated with Ciprofloxacin. However, the treatment of systemic anthrax patients was much more complicated with a failure rate of about 50%; five casualties out of 11 patients [17]. The treatment of symptomatic patients included the combination of three to five antibiotics and treatment failure was probably due to the choice of ineffective antibiotics such as cephalosporin, or due to delayed initiation of treatment [17]. In 2001 the CDC refined their recommendations for treating anthrax and included recommendations for PEP and symptomatic (systemic) patients [16]. For PEP the CDC recommended a 60 days treatment of Ciprofloxacin or Doxycycline, with the prolonged regimen designed to prevent late disease relapse from the spores deposited in the lung ("the spore depot" hypothesis).

For the treatment of symptomatic patients, the CDC recommended combined treatment of Ciprofloxacin or Doxycycline with one or two additional drugs, from the antibiotics listed, including among others clindamycin. The combination of clindamycin and Ciprofloxacin was shown to be effective in 2001. In an attempt to improve treatment efficacy at late stages of the progressed, systemic disease, the CDC acknowledged the risk of CNS infection and the need for specific and effective treatment with drugs that exhibit improved blood brain barrier (BBB) penetration [18]. An experts' committee convened by the CDC drafted recommendations for treating systemic anthrax in the presence and absence of indications for CNS infection [19]. These recommendations include carbapenems and Linezolid (oxazolidinone), relatively new antibiotic classes that are efficient in treating other common resistant bacterial infections [20]. Though never used to treat anthrax in humans, these substances were recommended in combination with a fluoroquinolone, for the treatment of anthrax, mainly in the late meningeal stages of the disease [18]. Previously, we adapted a direct central nervous system (CNS) infection model by direct injection of encapsulated vegetative *B. anthracis* bacteria into the cerebral spinal fluid (CSF) [12, 21]. This model, named hereafter "intra cisterna magna" (ICM) injection, simulates the late meningitic stages of the disease with a brain pathology similar to that observed in the inhalational exposure model [11, 12]. Rabbits infected by ICM injection succumbed within 12 to 24h. At time of death, systemic bacteria dissemination is observed. This was shown by us to be toxin independent [12]. Using this ICM infection model we

demonstrated that treatment with Meropenem or Linezolid fails to protect rabbits from anthrax-meningitis as a single treatment or in combination with Ciprofloxacin [21]. In addition, we showed that the failure of Meropenem was due to slow BBB penetration rather than low antibacterial activity [21].

The need for new generation antibiotics is the consequence of constant appearance of antibiotic resistant pathogens, a process that does not necessarily apply to *B. anthracis*. *B. anthracis* is genetically stable [22], probably since this bacterium spends most of its life cycle in its dormant spore stage. Furthermore, since during the course of the infection, the major infection is blood borne that later disseminates into organs, *B. anthracis* has little to no interaction with host microbiome, making horizontal genetic transfer extremely unlikely. A possible result of this is the fact that naturally occurring isolates encoding multi-drug resistance were as yet not documented [23, 24].

In this manuscript, we test the efficacy of treating anthrax meningitis with classical β-lactams as either a single treatment or β-lactams in combination with β-lactamase inhibitors. The efficacy of Ampicillin, Amoxicillin/clavulanate (Augmentin), Ampicillin/sulbactam (Unasyn) and Piperacillin/tazobactam (Tazocin) were tested in the rabbit CNS infection model.

## Material and methods

### Bacterial strains, media and growth conditions

The *B. anthracis* strain used in this study is the Vollum strain (ATCC14578) [25]. *B. anthracis* was cultivated in Terrific broth [26] at 37˚C. For the induction of toxins and capsule production, DMEM-10% NRS was used.

### Infection of rabbits

New Zealand white rabbits (2.5–3.5 kg) were obtained from Charles River (Canada). The animals received food and water ad libitum. Prior to infection, spores were germinated by incubation in Terrific broth for 1 hr, and then incubated in DMEM-10% NRS for 2 hr at 37˚C with 10% $CO_2$ to induce capsule formation. The capsule was visualized by negative staining with India ink. The encapsulated vegetative bacteria were used to infect rabbits via the intra-cisterna magna (ICM). For ICM administration, the animals were randomly divided into groups (as described in the results) and anesthetized using 100mg ketamine and 10mg xylazine. Using a 23 G blood collection set, 300 μl of encapsulated vegetative bacteria were injected into the cisterna magna. The remaining sample was plated for total viable counts (CFU.ml-1). The animals were observed daily for 14 days or for the indicated period. Upon death, blood and brain samples from selected rabbits were plated to determine presence of bacteria.

### Antibiotic treatment

Exposure and treatment regimens. Groups of 8 to 12 NZW rabbits (Charles River Canada. 2.5 to 3 kg) were inoculated ICM with $1x10^5$ Vollum encapsulated bacteria. Six hours post inoculation the animals were treated intravenously (IV) with antibiotics for a fast drug exposure. Antibiotic treatment continued twice daily subcutaneously (SC) for a period of 5 days, followed by survival monitoring for 9 additional days. Antibiotics doses were as follows: Meropenem 40–150 mg/kg, Ampicillin 150–300 mg/kg, Augmentin 150 mg/kg, Unasyn 150 mg/kg, Tazocin 150–300 mg/kg. MIC for Ampicillin, Amoxicillin or Amoxillin/clavulonic (Augmentin) is 0.023 μg/ml (by Etest), Meropenem 0.064 μg/ml (by Etest) and Piperacillin (Tazocin) 0.125 μg/ml (microdilution).

This study was carried out by trained personnel, in strict accordance with the recommendations of the Guide for the Care and Use of Laboratory Animals of the National Research Council. All protocols were approved by the "IIBR committee on the Ethics of Animal Experiments", permits no. RB-13-17, RB-06-18, RB-14-18. We used female rabbits in these experiments since there are not significant differences in *B. anthracis* pathogenicity between male and female rabbits [27]. Before ICM injection, animals were sedated using 100mg ketamine and 10mg xylazine. Animals were monitored twice a day and euthanized immediately when one of the following symptoms was detected: severe respiratory distress or the loss of righting reflex, by a 120 mg/kg sodium pentabarbitone injection. Animals that were unable or unwilling to drink were injected with 20-100ml of saline or dextrose isotonic solution SC. Since in most cases anthrax symptoms in rabbits are visible only in close proximity to death there were cases were animals succumbed to the disease.

## Statistical analysis

Statistical analysis. The significance of the differences in survival rates between treated groups and untreated controls and of the differences in bacteremia and time to death were determined by Log-rank, using Prism 6 software (Graphpad, USA).

## Results

### Efficacy of Ampicillin and Augmentin in treating anthrax CNS infections

To test the efficacy of Ampicillin and Augmentin (Amoxicillin/Clavulanate) we used our rabbit CNS infection model. Rabbits were infected by intra-cisterna magna (ICM) injection with capsular vegetative Vollum strain bacteria. Six hours post infection the rabbits were treated with 150 mg/kg Ampicillin or Augmentin by intravenous (IV) injection. The rabbits were then treated twice a day with 150 mg/kg subcutaneously (SC) for 5 additional days. Survival was monitored for a total of 14 days from infection. The results (**Fig 1**) demonstrate that treatment with Ampicillin protected more than 60% (8/12) of the treated rabbits. This is a significant improvement over Meropenem, that failed to protect any of the treated animals ([21] 0/8).

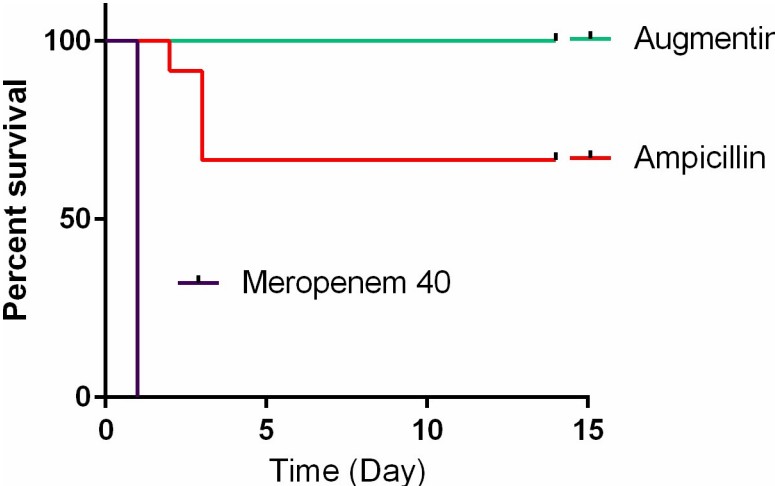

**Fig 1. Efficacy of treating anthrax CNS infection with Ampicillin or Augmentin.** Rabbits were inoculated ICM with vegetative encapsulated Vollum bacteria and treated 6 hr post infection with Meropenem (40 mg/kg), Ampicillin or Augmentin (150 mg/kg). Survival and time to death (Kaplan-Mayer curves) are shown. Statistical significance was P = 0.0001 for Meropenem vs. Ampicillin and P = 0.0779 for Ampicillin vs. Augmentin.

Though untreated animals or animals that are treated with ineffective drugs (Meropenem for example) die within 24h from infection, the time to death in the Ampicillin treated animal occurred between 48 to 72h post infection.

Treating the rabbits with Augmentin was highly effective (**Fig 1**) protecting all the animals (8/8) from the ICM infection with the Vollum fully virulent strain. The difference between Ampicillin and Augmentin could be the combination of the synergism in antimicrobial activity of Amoxicillin and Clavulanate or the inhibition of any β-lactamase activity.

## Effect of compromising the BBB on Ampicillin efficacy

Though the efficacy of Ampicillin was relatively high, we wanted to test whether it could be further improved. Following our previous experience, we tested if the limited efficacy of the Ampicillin treatment was due to slow BBB penetration. As previously demonstrated [21] we developed two experimental protocols to test this particular issue: the first bypasses the BBB by directly injecting the antibiotic into the cerebrospinal fluids (CSF). The second protocol is based on the fact that enhanced immune responses to CNS infections result in massive infiltration of immune cells from both the innate and adaptive systems [28, 29]. Therefore, vaccination with a PA based vaccine prior to ICM inoculation with a toxin-producing *B. anthracis* strain will result in increased cerebral immune responses to the inoculation causing enhanced BBB permeability. We have previously demonstrated that this vaccination does not protect rabbits from ICM infections [21]. In addition, by comparing ICM infections using either toxin-producing strains or mutants completely deficient in toxin production, we surmised that the disease progression is different, suggesting the vaccination does not result in efficient toxin neutralization [21]. Nevertheless, we cannot exclude minor effects of anti-toxin antibodies on the efficacy of the antibiotic treatment.

Supplementing the IV Ampicillin treatment with a single injection of 350 μl of 15 mg/ml ICM protected all the animals from ICM infection (8/8) (**Fig 2**). The same increase in efficacy was achieved by immunizing the animals prior to the ICM inoculation (8/8). These results demonstrate that any treatment that compromises or bypasses the BBB increases the efficacy of Ampicillin in this treatment model (**Fig 2**). On the other hand, increasing the IV Ampicillin

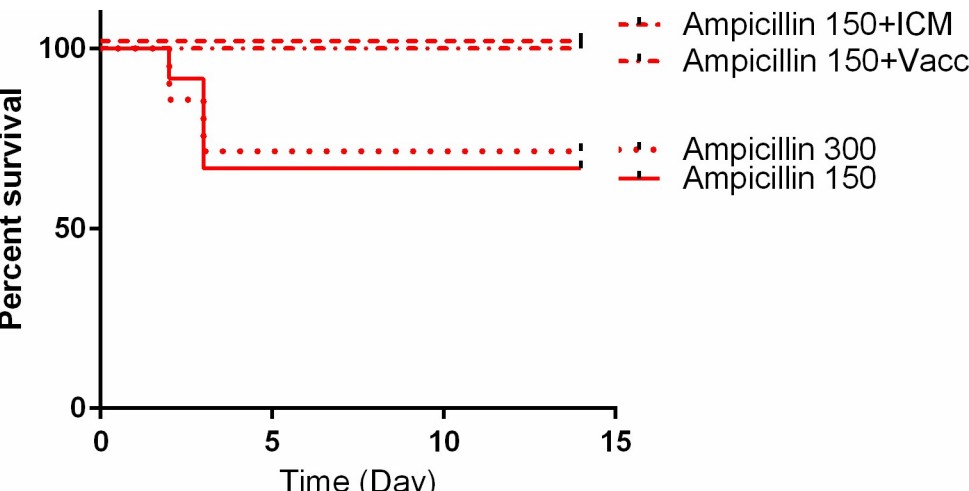

**Fig 2. Effect of compromising the BBB on the efficacy of Ampicillin treatment.** Naïve or pre-immune (Ampicillin 150+Vacc) rabbits were inoculated ICM with encapsulated Vollum bacteria and treated 6 hr post-infection with Ampicillin with the indicated dose (mg/kg) by IV or IV+ICM (Ampicillin 150+ICM). Survival and time to death are shown. Statistical significance P = 0.0779 for Ampicillin 150/300 vs. Ampicillin+ICM/PA.

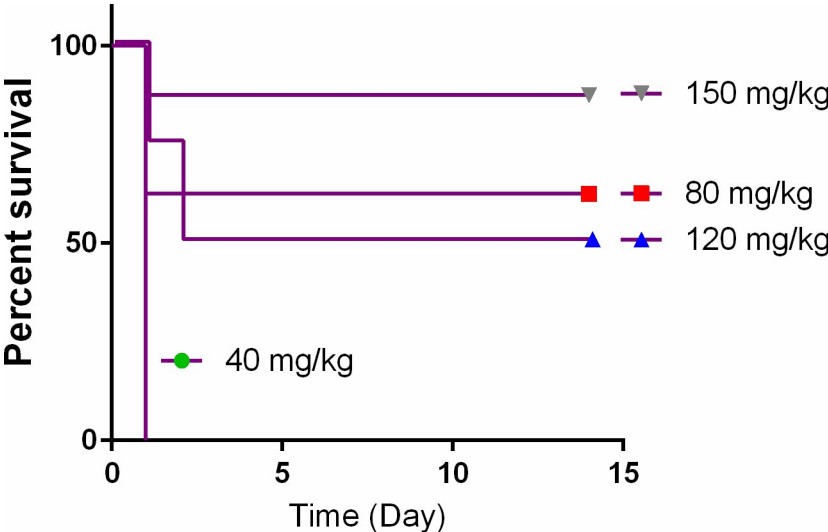

**Fig 3. Dose effect of Meropenem treatment or anthrax CNS infection.** Naïve rabbits were inoculated ICM with encapsulated Vollum bacteria and treated IV 6 hr post-infection with Meropenem at the indicated dose (mg/kg). Survival and time to death (curves) are shown. Statistical significance P = 0.0475 for Meropenem 40 mg/kg vs. 80 mg/kg and P = 0.26 for Meropenem 80 mg/kg vs. 150 mg/kg. Over all the results indicate a significant positive effect for increasing Meropenem doses beyond 40 mg/kg (P<0/0001).

dose by a factor of two from 150 to 300 mg/kg had no effect on treatment efficacy or time to death (0/7) (**Fig 2**).

## Effect of increasing Meropenem doses on treatment efficacy

Meropenem is an antibacterial very commonly used in hospitals and emergency rooms. Since our previous results indicated that the failure of the treatment was due to slow BBB penetration rather than low antibacterial activity, we tested the effect of increasing the therapeutic dose on the treatment efficacy. Increasing the dose of the two initial administrations of Meropenem from 40 to 80 mg/kg, followed by a return to 40 mg/kg for the rest of the treatment, increased the efficacy from 0 to over 50% (6/8) (**Fig 3**). Time to death in treatment failures remained unchanged with the animals succumbing within 24 hr from infection. Increasing the dose to 120 mg/kg (3 times the initial dose) did not achieve a significant increase in efficacy (2/4) compared to 80 mg/kg but the time to death was slightly prolonged to 48h. Finally, increasing the dose to ~four fold the initial dose to 150 mg/kg further increased efficacy to about 90% (7/8) (**Fig 3**), which we consider a fully protective treatment regimen. The effect of 40 mg/kg and 150 mg/kg Meropenem on the pharmacokinetics of this drug in rabbit serum is presented in S1 Apendix presenting effect on the C-max and T-min.

## Efficacy of Ampicillin/Sulbactam (Unasyn) or Piperacillin/Tazobactam (Tazocin) in treating CNS infections

Since Augmentin (Amoxicillin/Clavulanate) was highly effective in treating anthrax CNS infections (**Fig 1**), we tested the efficacy of additional β-lactam/β-lactamase-inhibitor combinations. We tested Unasyn and Tazocin in our rabbit CNS infection model. Treating the infected animals with 150 mg/kg of Unasyn protected ~90% of the animals (7/8) with a prolonged time to death for the animal that succumbed to infection under this treatment protocol (**Fig 4**). Treating the infected animals with 150 mg/kg Tazocin proved ineffective and did not

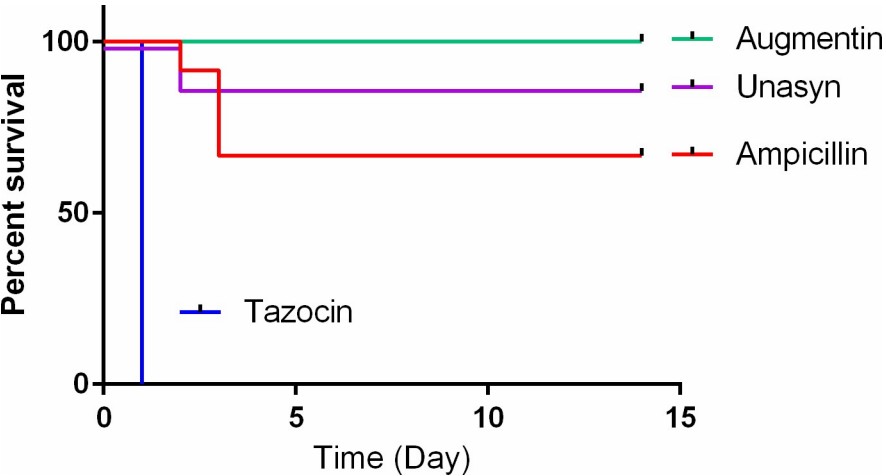

**Fig 4. Efficacy of Unasyn or Tazocin in treating anthrax CNS infections.** Rabbits were inoculated ICM with vegetative encapsulated Vollum bacteria and treated 6 hr post infection with Unasyn or Tazocin (150 mg/kg). The results of Ampicillin and Augmentin were copied from Fig 1. Survival and time to death are shown. Statistical significance P = 0.3173 for Augmentin vs. Unasyn and P = 0.0009 for Augmentin vs. Tazocin. Over all the results indicate a significant negative effect for the treatment of Tazocin compared to the other treatments (P<0/0001).

protect any of the animals treated (0/8). The time of death of the animals was not affected either, with all the animals dying within 24 hr (**Fig 4**).

We tried to improve the efficacy of the Tazocin treatment protocol by applying the BBB-permeability increasing methods described above, similarly to Meropenem and Ampicillin (**Fig 2**). We tested the effect of increasing the therapeutic dose from 150 mg/kg to 300 mg/kg, (n = 4) combining direct ICM treatment in combination with the 300 mg/kg treatment (n = 4), or pre-vaccination of the animals with a PA based vaccine (n = 4). None of these treatments had any effect on the treatment efficacy (**Fig 5**). The only effect observed was a

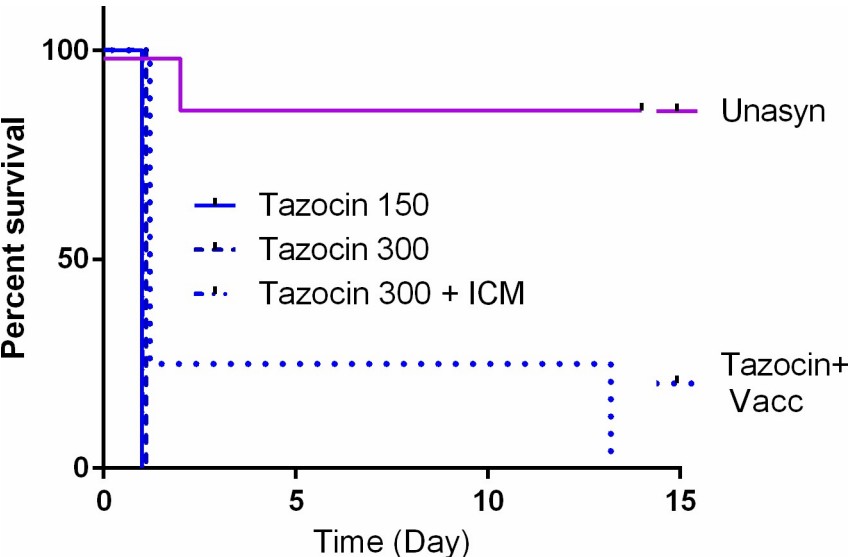

**Fig 5. Effect of compromising the BBB on the efficacy of Tazocin treatment.** Naïve or pre-immune (Tazocin 150 +Vacc) rabbits were inoculated ICM with encapsulated Vollum bacteria and treated 6 hr post-infection with Tazocin at the indicated dose (mg/kg) by IV or IV+ICM (Tazocin 300+ICM). The results of Unasyn were added from Fig 4. Survival and time to death are shown.

prolonged time to death for a single animal that was pre-vaccinated. This animal succumbed 13 days from infection while the rest of the animals in this experiment died within 24 hr from infection.

## Discussion

*B. anthracis*, unlike other tier 1 bacterial agents such as *Yersinia pestis* and *Francisella tulerensis* that are endemic and active in different parts of the world, has a very short infection cycle ending in the physiologically dormant spore state [30]. The time gap between natural infection cycles could be years or even decades, during which time the spores remain genetically stable. This is the reason why strains isolated from different parts of the world are almost identical [22] and could practically be genetically considered as different isolates, rather than strains. In addition, once exposed, the infection can either establish a disease that results in rapid death, or be cleared rapidly by the innate immune system, without the development of protective immunity [31]. *B. anthracis* does not form chronic infections, which usually enable the bacteria to develop antibiotic resistance. This could explain why naturally occurring antibiotic resistance in *B. anthracis* is very rare [23, 24]. Despite the relative genetic stability of *B. anthracis* and the lack of reported multiple drug resistances in naturally occurring isolates, the threat of using this bacterium as a bio-weapon raises the possibility of having designed resistances added to the dispersed spores. This, in combination with the additional possibility of a silent dispersal of spores in the case of a terror attack may result in delayed diagnosis and the development of severe, late stage disease in multiple patients, which is difficult to treat even in the case that antibiotic-sensitive strains are used. Therefore, new treatment protocols are sought after, using both existing and new antibacterial compounds. The chosen protocols will affect the types of antibiotics in national stockpiles as part of the preparedness for bio-terror attacks.

Previously we demonstrated that Linezolid and Meropenem are not effective in treating anthrax derived CNS infections although they are recommended by the CDC as first choice drugs for this stage of disease [21]. These drugs are generally used to treat common hospital infections and bacterial meningitis that are resistant to other, older compounds. In this manuscript, we tested the efficacy of β-lactams to treat anthrax-induced CNS infections in rabbits. Since antibiotic susceptibility testing of representative *B. anthracis* isolates from around the word revealed Ampicillin resistance in about 5% of the isolates [23] the use of this antibiotic was limited to strains that were pre-determined as sensitive. Furthermore, the genome of *B. anthracis* encodes β-lactamases that might cause this Ampicillin resistance [32]. Therefore, we also tested the combination of β-lactams with β-lactamase inhibitors.

Ampicillin, as a single antibiotic treatment, was effective in about 60% of the animals following ICM inoculation of the Vollum strain (**Fig 1**). The time to death of animals that succumbed to the infection was ~48 h longer than those succumbing while treated with Meropenem or while completely untreated ([12, 21] and **Fig 1**). This prolonged time to death could indicate the emergence of inducible resistance (β-lactamases). Attempts to improve the BBB penetration of Ampicillin by direct injection of the antibiotic into the CSF improved the treatment efficacy to 100%. This might indicate that rapid accumulation of Ampicillin in the CSF promptly kills the bacteria and negates their ability to induce resistance. The fact that a two fold increase in the Ampicillin dose did not improve the treatment efficacy implies that for effective treatment the Ampicillin dose should be higher still, raising the issue of safety under such high doses. However, Ampicillin treatment is significantly more effective than that of Meropenem (P = 0.0001). As we demonstrated previously, the low efficacy of Meropenem was due to slow BBB penetration [21]. Since Meropenem is considered safe, the option to dramatically increase doses is available. We therefore tested the efficacy of Meropenem with doses

as high as four times the initial usual dose used in rabbits. In this case (Fig 3) treating the rabbits with 150 mg/kg for the two initial doses followed by a return to 40 mg/kg protected about 90% of the animals, compared to 0% when the animals were treated with 40 mg/kg throughout the experiment (P = 0.005). Therefore, the recommendation of maximizing the initial dose must be included in the recommendation for using Meropenem for systemic anthrax with possible meningitis.

The presence of β-lactamase producing *B. anthracis* isolates limits the use of β-lactams such as Ampicillin and Amoxicillin in treating anthrax. However the vast majority of the tested strains were sensitive to Augmentin (Amoxicillin/Clavulanate) [23]. Indeed, Augmentin treatment was highly effective in treating CNS infected rabbits, protecting 100% of the animals (Fig 1). Similar results were obtained with Unasyn (Ampicillin/ Sulbactam). Unasyn treatment protected about 90% of the animals following CNS inoculation of *B. anthracis*. The time to death of the only animal not protected was 48 hr, which is 24 hr longer than that of the untreated animals [11, 12, 21]. The difference between Ampicillin and Augmentin or Unasyn could be the inhibition of the inducible β-lactamase present in the genome of *B. anthracis* or the synergism between the β-lactamase inhibitors (Clavulonate or Sulbactam), known to have antibacterial activity of their own [33]. In contrast to Augmentin and Unasyn, Tazocin (Piperacillin/Tazobactam) treatment did not protect any of the animals (Fig 4). This failure was probably due to low activity of Piperacillin rather than slow BBB penetration since compromising or bypassing the BBB by pre-vaccination or direct CSF administration did not improve the efficacy of this antibiotic (Fig 5). Piperacillin could be an additional example of a new generation drug that has lower efficacy in treating meningeal anthrax compared to the older drugs (Ampicillin and Amoxicillin)[34].

To conclude, in this manuscript we demonstrate that the use of Ampicillin or Amoxicillin in combination with the appropriate β-lactamase inhibitors is superior to the treatment based on Meropenem or Piperacillin. If using Meropenem, the initial treatment must be given at the maximal dose possible (4x the usual dose, according to the rabbit model). Tazocin has been shown to be ineffective, whereas Ampicillin and Amoxicillin exhibited high efficacy in the treatment of meningeal anthrax.

## Supporting information

**S1 Appendix. Pharmacokinetics of Meropenem following IV injection of 40 mg/kg or 150 mg/kg to rabbits.**
(DOCX)

## Author Contributions

**Conceptualization:** Assa Sittner, Amir Ben-Shmuel, Itai Glinert, David Kobiler, Shay Weiss, Haim Levy.

**Data curation:** Assa Sittner, Amir Ben-Shmuel, Itai Glinert, Elad Bar-David, Josef Schlomovitz, Shay Weiss, Haim Levy.

**Formal analysis:** Assa Sittner, Amir Ben-Shmuel, David Kobiler, Shay Weiss, Haim Levy.

**Investigation:** Assa Sittner, Amir Ben-Shmuel, Itai Glinert, Shay Weiss.

**Methodology:** Amir Ben-Shmuel, Itai Glinert, Elad Bar-David, Josef Schlomovitz, Haim Levy.

**Supervision:** Haim Levy.

**Writing – original draft:** Haim Levy.

**Writing – review & editing:** Assa Sittner, Amir Ben-Shmuel, Itai Glinert, Elad Bar-David, David Kobiler, Shay Weiss.

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
