## [Decision Letter · Decision Letter 0]

20 Dec 2019

PONE-D-19-30028

Using old antibiotics to treat ancient bacterium - β-lactams for Bacillus anthracis meningitis

PLOS ONE

Dear Dr. Levy,

Thank you for submitting your manuscript to PLOS ONE. After careful consideration, we feel that it has merit but does not fully meet PLOS ONE’s publication criteria as it currently stands. Therefore, we invite you to submit a revised version of the manuscript that addresses the points raised during the review process.

Reviewers have provided input on your manuscript and have recommended a major revision be completed. I invite you to review the comments provided, and if willing to undertake the work recommended, to provide a revised manuscript and a point-by-point response to each comment raised by the reviewers. Additional experiments to address the issue of PA vaccination to increase BBB penetration (particularly using VollumΔTox instead of the Vollum strain to infect the animals following PA vaccination) should be completed as recommended by Reviewer #2. Furthermore, more explanation regarding which stage of disease is being modeled in the animal studies needs to be addressed, as recommended by Reviewer #1. Please also review the order of the figures as there appears to be an issue regarding the order/numbering of the figures and remove panels which duplicate presentation of data (e.g. present only the survival curves or the bar graphs) in Figures 1-5.

We would appreciate receiving your revised manuscript by January 30, 2020. To enhance the reproducibility of your results, we recommend that if applicable you deposit your laboratory protocols in protocols.io, where a protocol can be assigned its own identifier (DOI) such that it can be cited independently in the future. For instructions see: http://journals.plos.org/plosone/s/submission-guidelines#loc-laboratory-protocols

We look forward to receiving your revised manuscript.

Kind regards,

Haroon Mohammad

Academic Editor

PLOS ONE

Journal Requirements:

Please ensure that your manuscript meets PLOS ONE's style requirements, including those for file naming. The PLOS ONE style templates can be found at http://www.plosone.org/attachments/PLOSOne_formatting_sample_main_body.pdf and http://www.plosone.org/attachments/PLOSOne_formatting_sample_title_authors_affiliations.pdf To comply with PLOS ONE submissions requirements, in your Methods section, please provide additional information on the animal research and ensure you have included details on (1) methods of sacrifice, (2) methods of anesthesia and/or analgesia, and (3) efforts to alleviate suffering. Thank you for including your ethics statement:  "This study was carried out in strict accordance with the recommendations of the Guide for the Care and Use of Laboratory Animals of the National Research Council. The protocols were approved by the Committee on the Ethics of Animal Experiments of the IIBR.Permit no. RB-13-17, RB-04-18, RB-14-18. ".   To comply with PLOS ONE submissions requirements, please provide the following information in the Methods section of the manuscript and in the “Ethics Statement” field of the submission form (via “Edit Submission”):  *  Please indicate whether an animal research ethics committee prospectively approved this research or granted a formal waiver of ethics approval.*  Please enter the name of your Institutional Animal Care and Use Committee (IACUC) or other relevant ethics board. Also include an approval number if one was obtained.*   If anesthesia, euthanasia, or any kind of animal sacrifice is part of the study, please include briefly in your statement which substances and/or methods were applied.For additional information about PLOS ONE submissions requirements for ethics oversight of animal work, please refer to http://journals.plos.org/plosone/s/submission-guidelines#loc-animal-research  For additional information about PLOS ONE submissions requirements for animal ethics, please refer to http://journals.plos.org/plosone/s/submission-guidelines#loc-animal-research  Once you have amended this/these statement(s) in the Methods section of the manuscript, please add the same text to the “Ethics Statement” field of the submission form (via “Edit Submission”).

Reviewers' comments:

Reviewer's Responses to Questions

**Comments to the Author**

1. Is the manuscript technically sound, and do the data support the conclusions?

Reviewer #1: Partly

Reviewer #2: Yes

2. Has the statistical analysis been performed appropriately and rigorously? 

Reviewer #1: Yes

Reviewer #2: I Don't Know

3. Have the authors made all data underlying the findings in their manuscript fully available?

Reviewer #1: No

Reviewer #2: Yes

4. Is the manuscript presented in an intelligible fashion and written in standard English?

Reviewer #1: Yes

Reviewer #2: Yes

5. Review Comments to the Author

Reviewer #1: The manuscript submitted by Sittner et al. entitled “Using old antibiotics to treat ancient bacterium-B-lactams for bacillus anthracis meningitis” discusses several interesting findings that could be beneficial for better understanding B. anthracis pathogenesis and anthrax disease. However, there are several important aspects from this reviewer’s perspective that must be addressed prior to being suitable for publication.

MAJOR POINTS

1. The discussion of any impacts of toxin components seems to be missing from the manuscript. The models being used here are interesting and are certainly useful, but more detailed description of how they reflect natural disease is warranted. For example, when introducing encapsulated bacteria into the ICM of the rabbits and then starting treatment 6 hours later, what stage of disease is being modeled here? If it is late stage disease, the lack of toxemia or damage associated with toxemia must at least be addressed as a potential shortfall of this model. As the authors are well aware, B. anthracis pathogenesis and subsequent anthrax disease is a highly complex scenario involving spores, bacilli, and toxins, thus more effort should be focused on placing this useful model in context of natural disease.

2. It is unclear to this reviewer after reading this manuscript how vaccination with PA can compromise the BBB. Please make this point clearer and provide references or other rationale, I apologize if I missed this point. Additionally, similar to point #1 above, the authors should comment about how the PA vaccine also changes the pathogenesis of the disease in their rabbit models. The PA vaccine is potent and it would definitely impact the disease model.

MINOR POINTS

1. Line 25, what evidence is being referred to suggest a rise for bio-terror risk associated with spores? I would recommend altering this, to reflect that B. anthracis spores are a proven risk.

2. Line 55, the authors should address the lethality associated with cases of GI anthrax, as it can cause more than “gut inflammation”.

3. Although a common acronym, please define BBB first time used in manuscript (sorry if I missed it).

4. Line 142, “rabbit” should be “rabbits”

5. The figures seem to be out of order?

6. The data presentation is duplicative, there is no benefit in presenting the data as bar graphs and as survival curves. Remove the bar graphs.

Reviewer #2: The data provided in this manuscript generally support your conclusion that the old antibiotics can be used to treat anthracis meningitis efficiently. Below please see the comments.

1. In your manuscript, the numbering of Figures 2, 4 and 5 is wrong. They should be corrected.

2. Augment and Unasyn but not Tazocin were inefficient to protect ICM infected animals in your study. You explained that this was due to low activity of Piperacillin. You should provide the MIC data for the antibiotics (Amoxicillin, Ampicillin, Piperacillin and Meropenem) used in your study.

3. You used PA vaccination to increase BBB penetration. Your previous study (Treating Anthrax-Induced Meningitis in Rabbits. Antimicrobial Agents and Chemotherapy, 2018) showed that the presence of anthrax toxin promotes animal death (compare the survival time of animals infected with Vollum vs. VollumΔTox), which may be partially reversed by anti-PA antibody induced by PA vaccination. You cannot exclude the possibility that the increased survival of animals receiving Ampicillin 150 + Vacc (Fig. 5 in your submission) is due to toxin neutralization. To better address that increased BBB penetration can increase the efficacy of Ampicillin, VollumΔTox instead of Vollum strain should be used to infect the animals following PA vaccination. Please address this concern by experiments and/or discussion.

6. PLOS authors have the option to publish the peer review history of their article (what does this mean?). If published, this will include your full peer review and any attached files.

Reviewer #1: No

Reviewer #2: No

---

## [Author Response · Author response to Decision Letter 0]

29 Dec 2019

PONE-D-19-30028

Using old antibiotics to treat ancient bacterium - β-lactams for Bacillus anthracis meningitis

PLOS ONE

Dear Dr. Haroon Mohammad,

Enclosed pleas find a point by point reply for the reviewers remarks. We would like to thank the reviewers for careful reading of the manuscript and making helpful remarks.

We added in a supplement a pharmacokinetics of two doses of Meropenem, 40 and 150 mg/kg in naïve rabbits. 

Hope you will find the revised version of this manuscript suitable for publication in PLOS one.

Yours

Haim Levy

The ethic statement was modified to:

This study was carried out by trained personnel, in strict accordance with the recommendations of the Guide for the Care and Use of Laboratory Animals of the National Research Council. All protocols were approved by the IIBR committee on the Ethics of Animal Experiments, permits no. RB-13-17, RB-06-18, RB-14-18. We used female rabbits in these experiments since there are not significant differences in B. anthracis pathogenicity between male and female rabbits [27]. Before ICM injection, animals were sedated using 100mg ketamine and 10mg xylazine. Animals were monitored twice a day and euthanized immediately when one of the following symptoms was detected: severe respiratory distress or the loss of righting reflex, by the 120 mg/kg sodium pentabarbitone injection. Animals that were unable or unwilling to drink were injected with 20-100ml of saline or dextrose isotonic solution SC. Since in most cases anthrax symptoms in rabbit are visible only in close proximity to death there were cases were animals succumbed to the disease.

Reviewer #1: 

MAJOR POINTS

1. The discussion of any impacts of toxin components seems to be missing from the manuscript. The models being used here are interesting and are certainly useful, but more detailed description of how they reflect natural disease is warranted. For example, when introducing encapsulated bacteria into the ICM of the rabbits and then starting treatment 6 hours later, what stage of disease is being modeled here? If it is late stage disease, the lack of toxemia or damage associated with toxemia must at least be addressed as a potential shortfall of this model. As the authors are well aware, B. anthracis pathogenesis and subsequent anthrax disease is a highly complex scenario involving spores, bacilli, and toxins, thus more effort should be focused on placing this useful model in context of natural disease.

A. The following text was added to the introduction:

Previously, we adapted a direct central nervous system (CNS) infection model by direct injection of encapsulated vegetative B. anthracis bacteria into the cerebral spinal fluid (CSF) [12, 21]. This model, named hereafter “intra cisterna magna” (ICM) injection, simulates the late meningitic stages of the disease with brain pathology similar to that observed in the inhalational exposure model [11, 12]. Rabbits infected by ICM injection succumbed within 12 to 24h. At time of death, systemic bacteria dissemination is observed. This was shown by us to be toxin independent [12].

2. It is unclear to this reviewer after reading this manuscript how vaccination with PA can compromise the BBB. Please make this point clearer and provide references or other rationale, I apologize if I missed this point. Additionally, similar to point #1 above, the authors should comment about how the PA vaccine also changes the pathogenesis of the disease in their rabbit models. The PA vaccine is potent and it would definitely impact the disease model.

A. The following text was added to the results:

As previously demonstrated [21] we developed two experimental protocols to test this particular issue: the first bypasses the BBB by directly injecting the antibiotic into the cerebrospinal fluids (CSF). The second protocol is based on the fact that enhanced immune responses to CNS infections results in massive infiltration of immune cells from both the innate and adaptive systems [28, 29]. Therefore, vaccination with a PA based vaccine prior to ICM inoculation with a toxin-producing B. anthracis strain will result in increased cerebral immune responses to the inoculation causing enhanced BBB permeability.

MINOR POINTS

1. Line 25, what evidence is being referred to suggest a rise for bio-terror risk associated with spores? I would recommend altering this, to reflect that B. anthracis spores are a proven risk.

A. The text was modified to” As Bacillus anthracis spores pose a proven bio-terror risk,..”

2. Line 55, the authors should address the lethality associated with cases of GI anthrax, as it can cause more than “gut inflammation”.

A. The text was modified to “Both, the gastrointestinal infection, resulting from the consumption of contaminated meat causing gut inflammation [5] and inhalational anthrax, which is the result of spore inhalation, become systemic in almost 100% of the cases and are almost always lethal without treatment [3].” 

3. Although a common acronym, please define BBB first time used in manuscript (sorry if I missed it).

A. Corrected accordingly

4. Line 142, “rabbit” should be “rabbits”

A. Corrected accordingly

5. The figures seem to be out of order?

A. I apologize for that. The figures title were corrected.

6. The data presentation is duplicative, there is no benefit in presenting the data as bar graphs and as survival curves. Remove the bar graphs.

A. we accept the reviewer comment, however from our experience in presenting such data to non-experts, the bar presentation is easier to understand. If the reviewer will insist, we will remove the bars from the figures.

Reviewer #2: 

1. In your manuscript, the numbering of Figures 2, 4 and 5 is wrong. They should be corrected.

A. We apologize for that once again. The figure titles were corrected.

2. Augment and Unasyn but not Tazocin were inefficient to protect ICM infected animals in your study. You explained that this was due to low activity of Piperacillin. You should provide the MIC data for the antibiotics (Amoxicillin, Ampicillin, Piperacillin and Meropenem) used in your study.

A. The MIC of this data was presented in the material and methods section:

“Antibiotics doses were as follows: Meropenem 40-150 mg/kg, Ampicillin 150-300 mg/kg, Augmentin 150 mg/kg, Unasyn 150 mg/kg, Tazocin 150-300 mg/kg. MIC for Ampicillin, Amoxicillin or Amoxillin/clavulonic (Augmentin) is 0.023 �g/ml (by Etest), Meropenem 0.064 �g/ml (by Etest) and Piperacillin (Tazocin) 0.125 �g/ml (microdilution).”

In addition, we added the pharmacokinetics of Meropenem in two doses, 40 and 150 mg/kg in rabbits as a supplement.

3. You used PA vaccination to increase BBB penetration. Your previous study (Treating Anthrax-Induced Meningitis in Rabbits. Antimicrobial Agents and Chemotherapy, 2018) showed that the presence of anthrax toxin promotes animal death (compare the survival time of animals infected with Vollum vs. VollumΔTox), which may be partially reversed by anti-PA antibody induced by PA vaccination. You cannot exclude the possibility that the increased survival of animals receiving Ampicillin 150 + Vacc (Fig. 5 in your submission) is due to toxin neutralization. To better address that increased BBB penetration can increase the efficacy of Ampicillin, VollumΔTox instead of Vollum strain should be used to infect the animals following PA vaccination. Please address this concern by experiments and/or discussion.

A. This text was added to the results:

“We have previously demonstrated that this vaccination does not protect rabbits from ICM infections [21]. In addition, by comparing ICM infections using either toxin-producing strains or mutants completely deficient in toxin production, we surmised that the disease progression is different, suggesting the vaccination does not result in efficient toxin neutralization [21]. Nevertheless, we cannot exclude minor effects of anti-toxin antibodies on the efficacy of the antibiotic treatment.”

---

## [Decision Letter · Decision Letter 1]

15 Jan 2020

PONE-D-19-30028R1

Using old antibiotics to treat ancient bacterium - β-lactams for Bacillus anthracis meningitis

PLOS ONE

Dear Dr. Levy,

Thank you for submitting your manuscript to PLOS ONE. After careful consideration, we feel that it has merit but does not fully meet PLOS ONE’s publication criteria as it currently stands. Therefore, we invite you to submit a revised version of the manuscript that addresses the points raised during the review process.

Thank you for addressing the reviewers comments. After carefully reviewing your revised manuscript, I am requesting that the figures be modified to address a concern previously raised by both reviewers. Figures 1 - 5 should only be presented as survival curves (e.g. please remove all of the bar graphs). Additionally, for the supplementary figure, please correct the title by changing "adminiatration so" to "administration of" and include a brief explanation of the pharmacokinetic data for meropenem in the main manuscript text.

We would appreciate receiving your revised manuscript by January 27, 2020. To enhance the reproducibility of your results, we recommend that if applicable you deposit your laboratory protocols in protocols.io, where a protocol can be assigned its own identifier (DOI) such that it can be cited independently in the future. For instructions see: http://journals.plos.org/plosone/s/submission-guidelines#loc-laboratory-protocols

We look forward to receiving your revised manuscript.

Kind regards,

Haroon Mohammad

Academic Editor

PLOS ONE

Reviewers' comments:

Reviewer's Responses to Questions

**Comments to the Author**

1. If the authors have adequately addressed your comments raised in a previous round of review and you feel that this manuscript is now acceptable for publication, you may indicate that here to bypass the “Comments to the Author” section, enter your conflict of interest statement in the “Confidential to Editor” section, and submit your "Accept" recommendation.

Reviewer #1: (No Response)

Reviewer #2: All comments have been addressed

2. Is the manuscript technically sound, and do the data support the conclusions?

Reviewer #1: Yes

Reviewer #2: Yes

3. Has the statistical analysis been performed appropriately and rigorously? 

Reviewer #1: Yes

Reviewer #2: Yes

4. Have the authors made all data underlying the findings in their manuscript fully available?

Reviewer #1: Yes

Reviewer #2: Yes

5. Is the manuscript presented in an intelligible fashion and written in standard English?

Reviewer #1: Yes

Reviewer #2: Yes

6. Review Comments to the Author

Reviewer #1: Thank you for making the requested edits and additions. However, I still feel strongly that by presenting the exact same data in 2 different graph formats, it causes confusion and over states the amount of data presented. Between a line graph and the text the data should be able to be presented in a manner that all readers can understand. Please streamline the data presentation is only represented in one format and the text.

Reviewer #2: (No Response)

7. PLOS authors have the option to publish the peer review history of their article (what does this mean?). If published, this will include your full peer review and any attached files.

Reviewer #1: No

Reviewer #2: No

---

## [Author Response · Author response to Decision Letter 1]

21 Jan 2020

The figures were modified as requested

---

## [Editor Report · Decision Letter 2]

23 Jan 2020

PONE-D-19-30028R2

Using old antibiotics to treat ancient bacterium - β-lactams for Bacillus anthracis meningitis

PLOS ONE

Dear Dr. Levy,

Thank you for submitting your manuscript to PLOS ONE. After careful consideration, we feel that it has merit but does not fully meet PLOS ONE’s publication criteria as it currently stands. Therefore, we invite you to submit a revised version of the manuscript that addresses the points raised during the review process.

We would appreciate receiving your revised manuscript by January 28, 2020. To enhance the reproducibility of your results, we recommend that if applicable you deposit your laboratory protocols in protocols.io, where a protocol can be assigned its own identifier (DOI) such that it can be cited independently in the future. For instructions see: http://journals.plos.org/plosone/s/submission-guidelines#loc-laboratory-protocols

We look forward to receiving your revised manuscript.

Kind regards,

Haroon Mohammad

Academic Editor

PLOS ONE

Additional Editor Comments (if provided):

Dear Dr. Levy,

Thank you for responding to the reviewer's request to modify the figures by removing the bar graphs. While reviewing the revised manuscript, I noticed that the figure legends for Figures 1-5 still mention "Survival percentage (bars) and time to death (Kaplan-Mayer curves) are shown." I would appreciate if you can correct all figure legends to remove the statement regarding survival percentage (bars) being shown (as this was deleted in the revised manuscript figures).

---

## [Author Response · Author response to Decision Letter 2]

23 Jan 2020

PONE-D-19-30028

Using old antibiotics to treat ancient bacterium - β-lactams for Bacillus anthracis meningitis

PLOS ONE

Dear Dr. Haroon Mohammad,

Enclosed pleas find a final version of the manuscript. We would like to thank the reviewers for careful reading of the manuscript and making helpful remarks.

Hope you will find the revised version of this manuscript suitable for publication in PLOS one.

Yours

Haim Levy

---

## [Editor Report · Decision Letter 3]

28 Jan 2020

Using old antibiotics to treat ancient bacterium - β-lactams for Bacillus anthracis meningitis

PONE-D-19-30028R3

Dear Dr. Levy,

We are pleased to inform you that your manuscript has been judged scientifically suitable for publication and will be formally accepted for publication once it complies with all outstanding technical requirements.

With kind regards,

Haroon Mohammad

Academic Editor

PLOS ONE

---

## [Editor Report · Acceptance letter]

3 Feb 2020

PONE-D-19-30028R3 

Using old antibiotics to treat ancient bacterium - β-lactams for Bacillus anthracis meningitis 

Dear Dr. Levy:

I am pleased to inform you that your manuscript has been deemed suitable for publication in PLOS ONE. Congratulations! Your manuscript is now with our production department. 

With kind regards,

on behalf of

Dr. Haroon Mohammad 

Academic Editor

PLOS ONE